# An unexpected diversity of powdery mildew species infecting the Fabaceae in Australia

**Lisa A. Kelly**[1,2]*, **Buddhika A. Dahanayaka**[1], **Niloofar Vaghefi**[1,3], **Aftab Ahmad**[1], **Levente Kiss**[1]

**1** Centre for Crop Health, University of Southern Queensland, Toowoomba, Australia, **2** Department of Primary Industries, Queensland Government, Toowoomba, Australia, **3** School of Agriculture, Food and Ecosystem Sciences, Faculty of Science, The University of Melbourne, Parkville, Australia

* lisa.kelly@daf.qld.gov.au

## Abstract

The Fabaceae family has been reported to host more than fifty species of powdery mildew worldwide. Despite being commonly found on fabaceous hosts throughout Australia, the accurate identification of many powdery mildew species remains uncertain. The objective of this study was to identify powdery mildew species that naturally occur on fabaceous hosts in Australia and provide insight into those native and weedy species that may host crop pathogens and contribute to disease in cropping systems. The ribosomal DNA internal transcribed spacer (ITS) sequences and morphology of 34 fresh and 40 herbarium powdery mildew specimens infecting diverse Fabaceae species in Australia were characterised in this study. Altogether, a total of eleven powdery mildew species were identified from 51 Fabaceae species. *Podosphaera xanthii* was the most common powdery mildew in this study and was detected on 18 host species across ten genera. Ten species of *Erysiphe* were confirmed on 37 host species covering 17 host genera, with *E. diffusa* and *E.* cf. *trifoliorum* the most prevalent. This work provides the most comprehensive catalogue of powdery mildew species infecting legume hosts throughout Australia.

## Introduction

The Fabaceae, commonly known as legumes, are one of the largest and economically important plant families worldwide, comprised of more than 640 genera and 18,000 species [1]. Legumes are widely distributed across every continent except Antarctica, occurring in almost every habitat type [1], and are known to improve soil fertility through their symbiosis with nitrogen-fixing bacteria [2]. Grain legumes, known as pulses, and forage legumes provide an important food source to humans and animals worldwide [2]. Several pulses are economically important broad-acre, horticultural or pasture crops in Australia, including common bean (*Phaseolus vulgaris* L.), pea (*Pisum sativum* L.), faba bean (*Vicia faba* L.), vetch (*Vicia* spp.), mungbean

**Data availability statement:** All molecular data is freely available in GenBank (https://www.ncbi.nlm.nih.gov/genbank/). All specimens collected were deposited in the Brisbane Plant Pathology Herbarium. Herbarium and GenBank accession numbers are listed in Table 2 within the paper.

**Funding:** The author(s) received no specific funding for this work.

**Competing interests:** The authors have declared that no competing interests exist.

(*Vigna radiata* [L.] Wilczek), black gram (*V. mungo* [L.] Hepper), cowpea (*V. unguiculata* [L.] Walp.), adzuki bean (*V. angularis* [Willd.]), soybean (*Glycine max* [L.] Merr.), lupin (*Lupinus angustifolius* [L.]), pigeonpea (*Cajanus cajan* [L.]), and various medics (*Medicago* spp.). The inclusion of these legume crops in the agricultural system provides multiple benefits to growers when grown in rotation with cereals, such as minimizing the build-up of cereal pathogen populations, improving soil fertility and biodiversity, and providing an additional source of income [3]. Several introduced Fabaceae have become significant weeds throughout Australia [4]. One example is *Macroptilium lathyroides*, commonly known as phasey bean, which was originally introduced as a pasture legume but now commonly grows throughout urban and regional habitats [5].

The land flora of Australia is unique as it includes approximately 11,119 native species that have evolved largely in isolation during the past 100 million years, following the separation of the Australian continent from other lands [6–7]. More than twice the number of native plant species were introduced to Australia as crops, pasture species, ornamentals, and also inadvertently as weeds, since the beginning of European colonization of the continent in the late 1700s [6–7]. During the past 5–10,000 years, and possibly also earlier, new plant species may have entered Australia naturally, as well, through land bridges that were formed between the continent and some of the neighbouring lands due to low sea levels during ice ages [8]. Some legume species are considered as Australian natives while others, including all commercially grown pulses, have been introduced since the first European settlers arrived in Australia [6].

The full life cycle of most powdery mildews includes an asexual (anamorph) and a sexual (teleomorph) stage [9]. During the asexual life cycle, airborne conidia are produced in large numbers to allow the pathogen to spread across wider regions and infect the green tissues, and sometimes also other aerial tissues of their living host plants if available in the environment [10]. Many species are also capable of sexual reproduction, whereby hyphae of opposite mating types fuse, and produce the sexual fruiting bodies (the sexual morphs) known as chasmothecia. These contain ascospores and may survive for long periods of time without a living host [10–11]. Many powdery mildew species do not develop chasmothecia in tropical and subtropical regions where they can produce conidia on their hosts throughout the year [11].

The Fabaceae family has been reported to host more than fifty species of powdery mildew (Erysiphaceae, Helotiales) worldwide [9,12–15]. Powdery mildews have commonly been found on fabaceous hosts throughout Australia; however, the accurate identification of many historically collected specimens remains uncertain. To complicate the accurate identification of these species further, the sexual morphs of powdery mildews infecting species of the Fabaceae have never been reported in Australia [15–17]. Distinguishing species of powdery mildew belonging to the same genus, based solely on morphological characteristics of the asexual morph can be difficult [16]. Sequencing the internal transcribed spacer (ITS) region of the nuclear ribosomal DNA (nrDNA) is considered a reliable method for distinguishing species of powdery mildew [16]. To date, eight powdery mildew species, belonging to two

genera, *Podosphaera* and *Erysiphe*, have been identified on fabaceous hosts in Australia based on morphology and ITS sequences (Table 1) [15–23]. In an earlier study by Cunnington et al. [15], restriction fragment length polymorphisms (RFLPs) and sequencing of the ITS region had also revealed the presence of one more taxon, identified as *Oidium hardenbergiae* on *Hardenbergia* spp., in Australia (Table 1).

Powdery mildews cause economically important diseases in several legume crops in Australia. This disease is reported to reduce grain yields by 20% in soybean [24] and up to 40% in mungbean [25–26]. A recent study by Kelly et al. [17] revealed that *Glycine tabacina*, a legume species native to Australia, is a host of the soybean pathogen, *Erysiphe diffusa*, and may be a source of inoculum to soybean crops growing nearby, as well as a means of survival on a perennial host between soybean cropping seasons. This case study has indicated that other members of the Fabaceae family may also act as alternate hosts for powdery mildew pathogens infecting crops grown in Australia.

Another recent study has revealed that powdery mildew epidemics on mungbean and black gram are caused by two species in Australia, not just *P. xanthii* as previously indicated [20]. *Erysiphe vignae*, the other causal agent of mungbean and black gram powdery mildew, may have been present in Australian paddocks long before it was discovered by Kelly et

**Table 1. Powdery mildew species and their fabaceous hosts reported in Australia prior to this study.**

| Powdery mildew species | Host plant | References |
|---|---|---|
| *Erysiphe aquilegiae* | *Cassia fistula* | [16] |
| *Erysiphe* cf. *trifoliorum* | *Acacia orites* | [16] |
| | *Pisum sativum* | [16] |
| | *Vicia tetrasperma* | [16] |
| *Erysiphe* cf. *trifoliorum*[1] | *Lathyrus odoratus* | [15] |
| | *Lupinus angustifolius* | [15] |
| | *Medicago* sp. | [15] |
| | *Trifolium* spp. | [15] |
| | *Sesbania* spp. | [15] |
| | *Trigonella suavissima* | [15] |
| | *Vigna dasycarpa* | [15] |
| *E. diffusa* | Glycine max | [16,17,20,22] |
| | Glycine clandestina | [16] |
| | Glycine tabacina | [17] |
| *E. medicaginis* | *Medicago polymorpha* | [18] |
| *E. pisi* | *Pisum sativum* | [19] |
| *E. quercicola* | *Acacia mangium* | [21,23] |
| | *Acacia holosericea* | [21,23] |
| | *Acacia sophorae* | [21,23] |
| *E. vignae* | *Vigna radiata* | [20] |
| | *Vigna mungo* | [20] |
| *Oidium hardenbergiae*[1] | *Hardenbergia* spp. | [15] |
| *Podosphaera xanthii* | *V. radiata* | [16,20] |
| | *Vigna mungo* | [20] |
| | *V. radiata* ssp. *sublobata* | [20] |
| *P. xanthii*[1] | *Phaseolus lathyroides* | [15] |

[1]Species were identified based on restriction fragment length polymorphisms (RFLPs) and/or sequencing of the nrDNA ITS region by Cunnington et al. (2004).

al. [20]. Little is known about *E. vignae* in general, including its possible alternate hosts. This knowledge gap triggered the current study that aimed at identifying powdery mildew species that naturally occur on fabaceous hosts in Australia based on nrDNA ITS sequences and/or morphological characteristics. The finding of alternate hosts of *E. vignae* and other crop pathogens may provide insight into the origin of diverse pathogens in cropping systems. This work provides the most comprehensive catalogue of powdery mildew species infecting legume hosts throughout Australia.

## Materials and methods

### Collection of powdery mildew specimens

Leaves of wild and cultivated legumes with naturally occurring powdery mildew infections were collected during *ad hoc* surveys throughout Queensland between 2020 and 2024. Specimens were also collected from other states in Australia, when possible. Most collections occurred at sites near cropping paddocks, where powdery mildew pathogens are most likely to occur. No materials were collected from natural settings that require a collection permit. Powdery mildew-infected tissues were microscopically examined and processed for DNA extraction as described below, prior to being dried and pressed as herbarium specimens and deposited at the Queensland Plant Pathology Herbarium (BRIP). Historically collected powdery mildew specimens from fabaceous hosts were also obtained from the Queensland Plant Pathology Herbarium and examined using light microscopy and ITS sequencing in order to identify the species based on the most recent taxonomic methods.

### Microscopic examination

Microscopic examination was performed using a Nikon Eclipse Ni-U (Tokyo, Japan) microscope with bright field and differential interference contrast (DIC) optics and photographs were taken with an Olympus DP23-CU 6.4MP (Tokyo, Japan) microscope camera. For fresh samples, actively growing mycelia, conidia and conidiophores were removed with cellotape and mounted on a microscope slide containing a droplet of lactic acid. Powdery mildew mycelia from infected herbarium specimens were rehydrated by boiling pieces of infected plant tissues in lactic acid on a microscope slide, as described by Shin and La [27]. During microscopy, the following characteristics were examined: shape and size of conidia (n = 25); presence or absence of fibrosin bodies in fresh conidia; nature of conidiogenesis; morphology of the conidiophore; shape of hyphal appressoria; and when observed, position of conidial germ tubes and shape of germ tube apices.

### DNA extraction, PCR amplification and sequencing

Powdery mildew mycelia were removed from fresh and herbarium infected plant tissues using 1–1.5 cm$^2$ pieces of cellotape, then total genomic DNA was extracted using an Extract-N-Amp Plant PCR kit (Sigma-Aldrich, St. Louis, MO) as per the manufacturer's instructions. Two DNA samples were prepared from each specimen. The nested polymerase chain reaction (PCR) method developed by Cunnington et al. [19], using primers PMITS1 and PMITS2 first, and ITS1-F and ITS4 primers in the second reaction, was modified to amplify the ITS region from the DNA samples of this study as described by Kiss et al. [16]. PCR products were purified and each strand was sequenced by Macrogen Inc. (Seoul, Korea) using primers ITS1-F and ITS4, respectively.

### Phylogenetic analyses

The ITS sequences of all specimens were edited using Geneious Prime 2024.0.5 (Dotmatics). Chromatograms were visually inspected for potential sequencing errors, trimmed and then the forward and reverse sequences were assembled to produce consensus sequences. These were used as queries in BLASTn searches in standard databases in NCBI GenBank and also in BLASTn searches that were limited to sequences from type materials. The consensus sequences

**Table 2. Powdery mildew specimens examined in this study and/or used in phylogenetic analyses.**

| Powdery mildew species | Host plant[1,2,3,4] | Herbarium accession number | Place and date of collection | ITS GenBank accession number | References |
|---|---|---|---|---|---|
| Cystotheca wrightii | Quercus glauca | MUMH 137 | Japan | AB000932 | [30] |
| Erysiphe alphitoides | Quercus sp. | VPRI 8763 | VIC | AB292705 | [35] |
| E. aquilegiae | Arachis pintoi[3,4] | BRIP 39239 | Mareeba, QLD | PQ509336 | This paper |
| | Cassia fistula[3] | BRIP 72798 | Australia, 2018 | PQ509337 | This paper |
| | C. fistula | BRIP 68834 | Toowoomba, QLD, 2018 | MT174205 | [16] |
| E. cf. trifoliorum | Acacia orites[1] | BRIP 70580 | Canungra, QLD, 2019 | MT174201 | [16] |
| | Arachis pintoi[3,4] | BRIP 47238 | Brisbane, QLD, 2005 | PQ509338 | This paper |
| | A. pintoi[3,4] | BRIP 25196 | Wamuran, QLD, 1998 | PQ509339 | This paper |
| | Medicago lupulina[3,4] | BRIP 20602 | Brisbane, QLD, 1992 | PQ509340 | This paper |
| | Phaseolus vulgaris[3,4] | BRIP 2236 | Glastonbury, QLD, 1970 | PQ509341 | This paper |
| | P. vulgaris[3,4] | BRIP 40189 | Bowen, QLD, 2003 | PQ509342 | This paper |
| | Pisum sativum[2] | BRIP 76640 | Toowoomba, QLD, 2022 | PQ509343 | This paper |
| | P. sativum | BRIP 68831 | QLD, 2017 | MT174202 | [16] |
| | Trifolium dubium[3,4] | BRIP 70118 | Coomera, QLD, 2018 | PQ509344 | This paper |
| | Vicia faba[2,4] | BRIP 74256 | Pampas, QLD, 2020 | OL966448 | This paper |
| | V. faba[2,4] | BRIP 74257 | Condamine Plains, QLD, 2020 | OL966449 | This paper |
| | V. faba[2,4] | BRIP 74258 | Condamine Plains, QLD, 2020 | OL966450 | This paper |
| | V. faba[2,4] | BRIP 74259 | Condamine Plains, QLD, 2020 | OL966451 | This paper |
| | V. faba[2,4] | BRIP 74260 | Formartin, QLD, 2020 | OL966452 | This paper |
| | V. sativa[2,4] | BRIP 76629 | Cecil Plains, QLD, 2019 | PQ509345 | This paper |
| | V. sativa[2,4] | BRIP 76634 | Pampas, QLD, 2021 | PQ509346 | This paper |
| | V. tetrasperma | BRIP 68838 | Tipton, QLD, 2017 | MT174203 | [16] |
| | V. villosa[2,4] | BRIP 76641 | Khosh Bulduk, QLD, 2021 | PQ509347 | This paper |
| E. diffusa | Acacia flavescens[1,3,4] | BRIP 34412 | Wongi, QLD | PQ509348 | This paper |
| | Cajanus cajan[2,4] | BRIP 76646 | Toowoomba, QLD, 2023 | PQ509349 | This paper |
| | C. cajan[3,4] | BRIP 64873 | Walkamin, QLD, 2016 | PQ509350 | This paper |
| | Clitoria sp.[3,4] | BRIP 11641 | Parada, QLD, 1976 | PQ509351 | This paper |
| | Cullen australasicum[1,2,4] | BRIP 76638 | Toowoomba, QLD, 2021 | PQ509352 | This paper |
| | Glycine clandestina[1] | BRIP 68827 | Toowoomba, QLD, 2018 | MT174188 | [16] |
| | G. max[2] | BRIP 76637 | Gatton, QLD, 2020 | PQ509353 | This paper |
| | G. max[2] | BRIP 76643 | Mackay, QLD, 2023 | PQ509354 | This paper |
| | G. max[3] | BRIP 55388 | Bowenville, QLD, 2012 | PQ509355 | This paper |
| | G. max[3] | BRIP 58077 | Bundaberg, QLD, 2012 | PQ509356 | This paper |
| | G. max[3] | BRIP 58459 | Kingaroy, QLD, 2013 | PQ509357 | This paper |
| | G. max[3] | BRIP 62030 | Ayr, QLD, 2014 | MT174190 | [16] |
| | G. tabacina[1] | BRIP 76160 | Toowoomba, QLD, 2021 | PP023535 | [17] |
| | Macrotyloma axillare x uniflorum[3,4] | BRIP 12960 | Walkamin, QLD, 1979 | PQ509358 | This paper |
| | M. daltonii[3,4] | BRIP 13833 | Brisbane, QLD, 1982 | PQ509359 | This paper |
| | M. uniflorum x africanum[3,4] | BRIP 12956 | Walkamin, QLD, 1979 | PQ509360 | This paper |
| | Phaseolus vulgaris[3,4] | BRIP 48539 | Ayr, QLD, 2004 | PQ509361 | This paper |
| E. glycines | Glycine max | MUMH 1462 | Japan | AB078807 | [29] |
| E. guarinonii | Baptisa australis | WTU-F-07242448 | USA, 2018 | MT516325 | [12] |

*(Continued)*

**Table 2.** (Continued)

| Powdery mildew species | Host plant[1,2,3,4] | Herbarium accession number | Place and date of collection | ITS GenBank accession number | References |
|---|---|---|---|---|---|
| | *Gastrolobium celsianum*[1,2,4] | BRIP 76684 | Parkville, VIC, 2022 | PQ509362 | This paper |
| | *Hardenbergia violacea*[1,2,4] | BRIP 76685 | Cranley, QLD, 2022 | PQ509363 | This paper |
| *E. medicaginis* | *Astragalus hamosus*[3,4] | BRIP 48604 | Toowoomba, QLD, 2006 | PQ509364 | This paper |
| | *Cullen tenax*[1,2,4] | BRIP 76633 | Pampas, QLD, 2020 | PQ509365 | This paper |
| | *C. tenax*[1,2,4] | BRIP 76632 | Mullaley, NSW, 2021 | PQ509366 | This paper |
| | *Hardenbergia comptoniana*[1,2,4] | BRIP 76683 | Fremantle, WA, 2022 | PQ509367 | This paper |
| | *Medicago littoralis*[3,4] | BRIP 48605 | Toowoomba, QLD, 2006 | PQ509368 | This paper |
| | *M. polymorpha* | BRIP 70957 | Toowoomba, QLD, 2019 | MT160214 | [18] |
| | *M. polymorpha*[2] | BRIP 76635 | Urrbrae, SA, 2020 | PQ509369 | This paper |
| | *M. truncatula*[2,4] | BRIP 76636 | Urrbrae, SA, 2020 | PQ509370 | This paper |
| *E. neolycopersici* | *Lupinus angustifolius*[3,4] | BRIP 8407 | Taabinga, QLD, 1974 | PQ509371 | This paper |
| | *Lycopersicon esculentum* | MUMH 66 | Japan, 1995 | LC009912 | [36] |
| | *Senna tora*[3,4] | BRIP 57411 | Cooktown, QLD, 2012 | PQ509372 | This paper |
| *E. pisi* | *Pisum sativum*[2] | BRIP 76639 | Warwick, QLD, 2022 | PQ509373 | This paper |
| | *P. sativum*[2] | BRIP 76644 | Hendon, QLD, 2023 | PQ509374 | This paper |
| | *P. sativum*[2] | BRIP 76645 | Hendon, QLD, 2023 | PQ509375 | This paper |
| | *P. sativum* | VPRI 19688 | Hopetoun, VIC, 1993 | AFO73348 | [19] |
| *E. quercicola* | *Acacia holosericea*[1] | VPRI 20468 | Humpty Doo, NT, 1995 | AB237806 | [21] |
| | *A. mangium*[1] | VPRI 20374 | Tully, QLD, 1994 | AB237807 | [21] |
| | *A. sophorae*[1] | BRIP 71600 | Tullymorgan, NSW, 2020 | MW293874 | [23] |
| | *Acacia* sp.[1,2,4] | BRIP 76686 | Crows Nest, QLD, 2021 | PQ509376 | This paper |
| *Erysiphe* sp. | *A. glaucoptera*[1,2,4] | BRIP 76682 | Perth, WA, 2022 | PQ509335 | This paper |
| *E. vignae* | *Macrotyloma axillare*[2,4] | BRIP 75422 | Brisbane, QLD, 2022 | PQ509377 | This paper |
| | *Vigna mungo* | BRIP 71598 | Toowoomba, QLD, 2019 | MW293895 | [20] |
| | *V. radiata* | BRIP 68837 | Toowoomba, QLD, 2018 | MT628284 | [20] |
| | *V. radiata* | BRIP 71005 | Toowoomba, QLD, 2019 | MT628282 | [20] |
| | *V. radiata* | BRIP 71006 | Warwick, QLD, 2019 | MT628285 | [20] |
| | *V. radiata* | BRIP 71010 | Gatton, QLD, 2019 | MT628286 | [20] |
| *Podosphaera aphanis* | *Fragaria* x *ananassa* | VPRI 19031 | VIC, 1993 | AF073355 | [19] |
| *P. fusca* | *Calendula officinalis* | VPRI 20625 | VIC, 1995 | AF154324 | [19] |
| *P. leucotricha* | *Maus domestica* | VPRI 17729 | NSW, 1991 | AF073353 | [19] |
| *P. pannosa* | *Rosa* sp. | BRIP 68844 | QLD, 2017 | MT174222 | [16] |
| *P. plantaginis* | *Plantago lanceolata* | BRIP 68845 | QLD, 2018 | MT174223 | [16] |
| *P. tridactyla* | *Prunus persica* | VPRI 19591 | VIC, 1982 | AY833653 | [19] |
| *P. tridactyla* | *Prunus* sp. | VPRI 19006 | VIC, 1993 | AF154321 | [19] |
| *P. xanthii* | *Alysicarpus rugosus*[1,3,4] | BRIP 13360 | Walkamin, QLD, 1981 | PQ509380 | This paper |
| | *Crotalaria lanceolata*[3,4] | BRIP 2215 | Calavos, QLD, 1968 | PQ509381 | This paper |
| | *C. mucronata*[3,4] | BRIP 2216 | Brisbane, QLD, 1966 | PQ509382 | This paper |
| | *Glycine max*[3,4] | BRIP 21968 | Gatton, QLD, 1994 | PQ509383 | This paper |
| | *Indigofera* sp.[3,4] | BRIP 8322 | Brisbane, QLD, 1962 | PQ509384 | This paper |
| | *Lathyrus odoratus*[3,4] | BRIP 2227 | Brisbane, QLD, 1969 | PQ509385 | This paper |
| | *Macroptilium atropurpureum*[3,4] | BRIP 71601 | Brisbane, QLD, 2020 | PQ509386 | This paper |
| | *M. lathyroides*[2,4] | BRIP 76642 | Cambooya, QLD, 2023 | PQ509387 | This paper |

*(Continued)*

**Table 2.** (Continued)

| Powdery mildew species | Host plant[1,2,3,4] | Herbarium accession number | Place and date of collection | ITS GenBank accession number | References |
|---|---|---|---|---|---|
| | *M. lathyroides*[3,4] | BRIP 71000 | Kalbar, QLD, 2019 | PQ509388 | This paper |
| | *Macrotyloma daltonii*[3,4] | BRIP 11696 | Walkamin, QLD, 1976 | PQ509389 | This paper |
| | *M. uniflorum*[3,4] | BRIP 11698 | Walkamin, QLD, 1976 | PQ509390 | This paper |
| | *Phaseolus vulgaris*[3,4] | BRIP 45128 | Bowen, QLD, 2004 | PQ509391 | This paper |
| | *P. vulgaris*[3,4] | BRIP 8404 | Maidenwell, QLD, 1979 | PQ509392 | This paper |
| | *P. vulgaris*[3,4] | BRIP 20729 | Brisbane, QLD, 1983 | PQ509393 | This paper |
| | *P. vulgaris*[3,4] | BRIP 25796 | Kurumba, QLD, 1999 | PQ509394 | This paper |
| | *Pisum sativum*[3,4] | BRIP 2239 | Brisbane, QLD, 1928 | PQ509395 | This paper |
| | *Vigna angularis*[2,4] | BRIP 76647 | Kingaroy, QLD, 2023 | PQ509396 | This paper |
| | *V. angularis*[3,4] | BRIP 21990 | Wooroolin, QLD, 1994 | PQ509397 | This paper |
| | *V. angularis*[3,4] | BRIP 23226 | Kingaroy, QLD, 1996 | PQ509398 | This paper |
| | *V. lanceolota*[1,2,4] | BRIP 76626 | Tummaville, QLD, 2024 | PQ509399 | This paper |
| | *V. lanceolota*[1,2,4] | BRIP 76625 | Pampas, QLD, 2021 | PQ509400 | This paper |
| | *V. mungo* | BRIP 71594 | Brisbane, QLD, 2019 | PQ509401 | [20] |
| | *V. radiata* | BRIP 71599 | Killarney, QLD, 2017 | MW293885 | [20] |
| | *V. radiata* ssp. *sublobata*[1,2] | BRIP 76630 | Clermont, QLD, 2021 | PQ509402 | This paper |
| | *V. radiata* ssp. *sublobata*[1,2] | BRIP 76631 | Clermont, QLD, 2021 | PQ509403 | This paper |
| | *V. unguiculata*[2,4] | BRIP 76627 | Warwick, QLD, 2020 | PQ509404 | This paper |
| | *V. unguiculata*[2,4] | BRIP 76628 | Warwick, QLD, 2020 | PQ509405 | This paper |
| | *V. unguiculata*[3,4] | BRIP 71439 | Brisbane, QLD, 2020 | PQ509406 | This paper |

[1]Native to Australia.

[2]Fresh specimen.

[3]Herbarium specimen.

[4]New host record in Australia.

produced in this study were deposited in GenBank (Table 2). Alignments of ITS sequences belonging to the *Podosphaera* and *Erysiphe* genera were constructed separately using MAFFT v. 7.388 [28], visually inspected for potential misalignments, then trimmed to the length of the shortest sequence. An *Erysiphe* ITS data set was constructed, consisting of 48 sequences from this study, 17 previously published sequences from Fabaceae hosts in Australia, and four sequences of representative specimens obtained from GenBank (Table 2). *Erysiphe glycines* MUMH 1462 (AB078807) was used as the outgroup in this alignment based on Takamatsu et al. [29]. This resulted in an alignment with a total length of 557 characters, including 417 identical and 140 variable sites.

A separate data set was constructed for specimens belonging to the *Podosphaera* genera. This dataset consisted of 26 sequences obtained in this study, two sequences from powdery mildew on *Vigna* spp. [20], and eight sequences of representative specimens obtained from GenBank (Table 2). *Cystotheca wrightii* MUMH 137 (AB000932) was used as the outgroup based on Takamatsu et al. [30]. This resulted in an alignment with a total length of 488 characters, including 363 identical and 125 variable sites.

Phylogenetic analyses were conducted separately on *Erysiphe* and *Podosphaera* ITS datasets using Bayesian Inference and Maximum Likelihood approaches. The Akaike Information Criterion was estimated for Bayesian Inference using MrModeltest v. 2.3. [31] and PAUP v. 4.0 [32] and used to determine the best-fit nucleotide substitution model for each alignment. Two Markov Chain Monte Carlo chains were run using MrBayes v. 3.2.4 [33], where one tree per 100 generations was saved, and the runs were ended when the standard deviation of split frequencies was below 0.01.

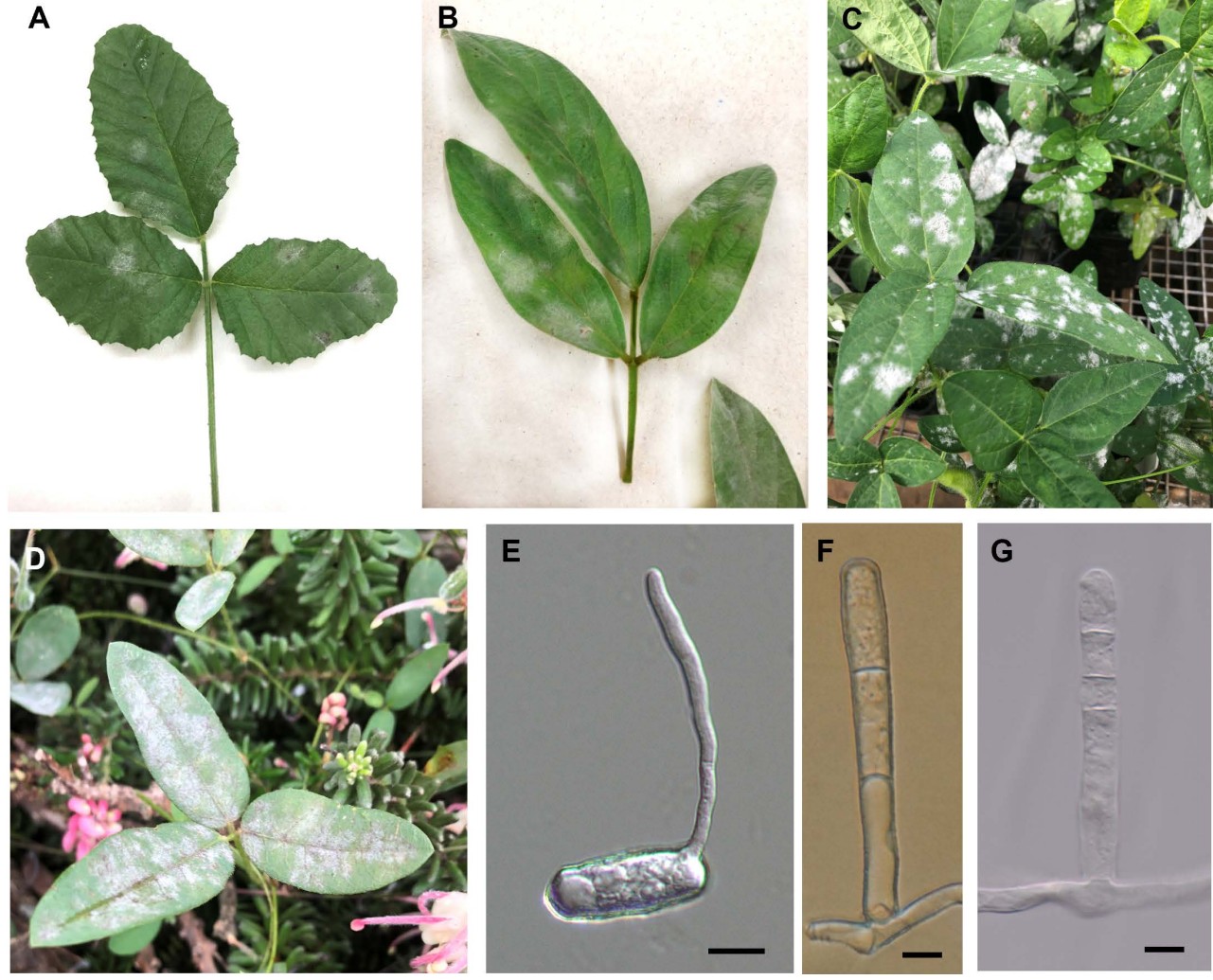

**Fig 1. Fabaceous hosts infected with *Erysiphe diffusa* in Australia.** (A) Symptoms on leaves of *Cullen australasicum* BRIP 76638, (B) *Cajanus cajan* BRIP 76646, (C) *Glycine max* BRIP 76637, and (D) *Glycine tabacina* BRIP 76160. (E) Germinated conidium mounted in lactic acid. (F - G) Conidiophores mounted in lactic acid. Bar = 10 μm.

The 50% majority rule consensus tree was estimated after a 25% burn-in of the saved trees. RaxML v. 8.2.11 [34] was used with the GTRGAMMA model of nucleotide substitution and 1,000 bootstrap replicates for the Maximum Likelihood analysis.

## Results

### Fabaceous hosts of powdery mildews in Australia

A total of 34 fresh powdery mildew specimens were collected in this study from 21 fabaceous host species across Australia (Table 2). An additional 40 herbarium specimens, from 27 host species, dating back to 1928, were also examined. Of all the hosts, the highest number of powdery mildew specimens were collected from the genus *Vigna*, with 17 specimens collected on six *Vigna* species (Table 2). The next largest group were powdery mildews on *Glycine* and *Vicia* species, with nine specimens of each. When combined with recently published specimens with available ITS sequences, this study

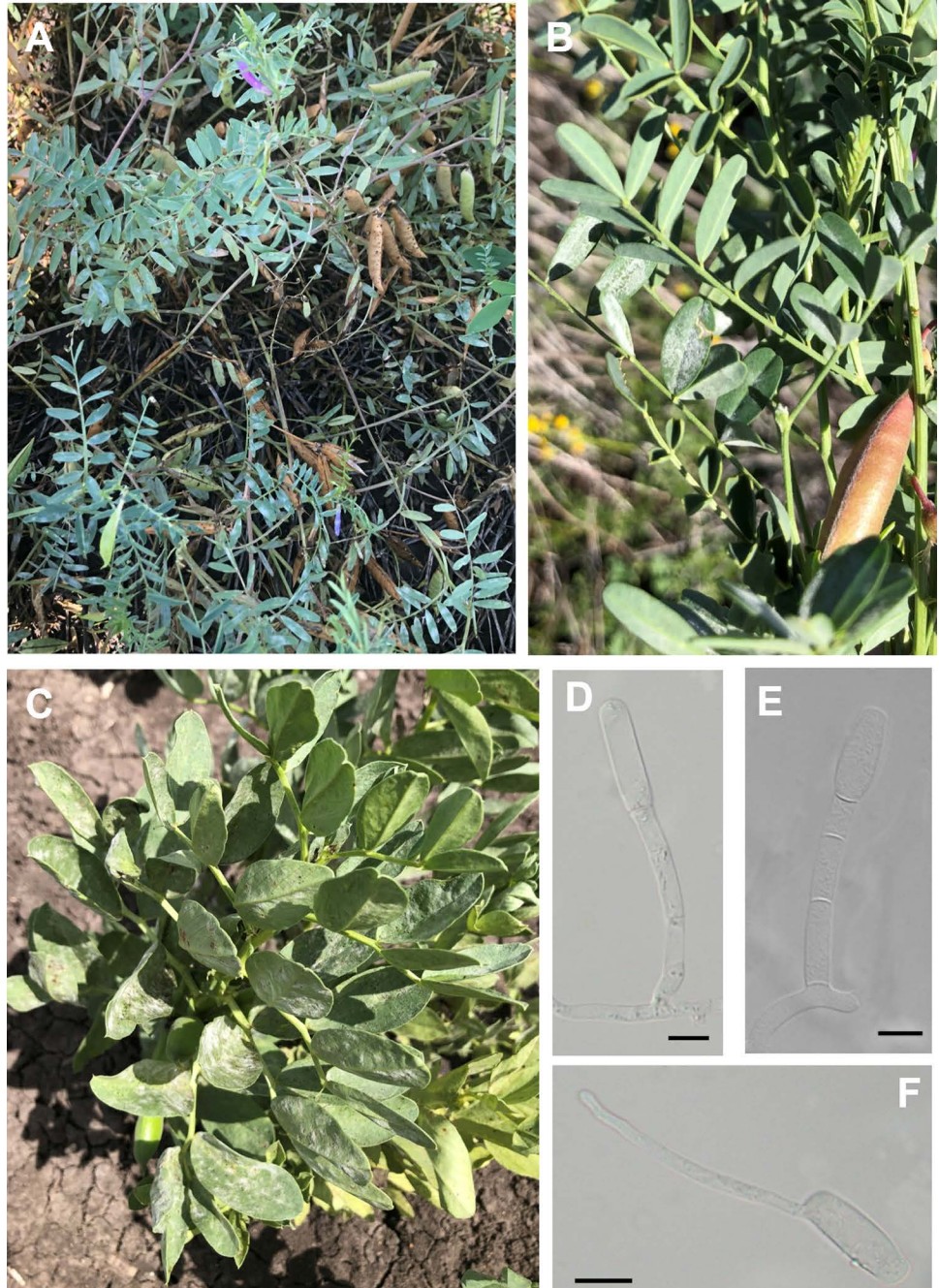

**Fig 2. Hosts in the Fabaceae infected with _Erysiphe_ cf. _trifoliorum_ in Australia.** (A) Symptoms on leaves of _Vicia sativa_ BRIP 76629, (B) _Vicia villosa_ BRIP 76641, and (C) _Vicia faba_ BRIP 74256. (D - E) Conidiophores mounted in lactic acid. (F) Germinated conidium mounted in lactic acid. Bar = 10 µm.

included 93 specimens on the Fabaceae, including powdery mildews from 17 native Australian host species and 34 hosts that were introduced to the continent from overseas (Table 2). All specimens were collected from Queensland (QLD) except for seven from Victoria (VIC); two each from Western Australia (WA), New South Wales (NSW), and South Australia (SA); and one from the Northern Territory (NT) (Table 2).

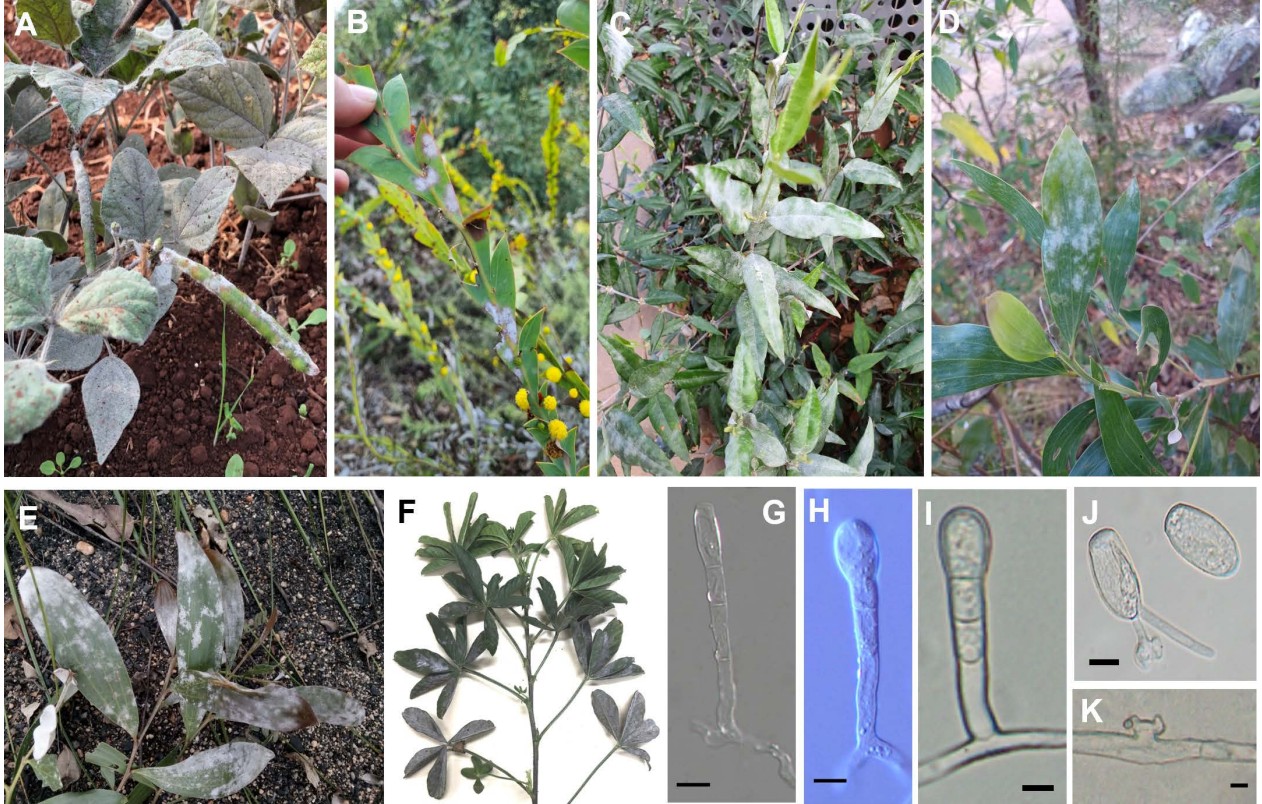

**Fig 3. Hosts in the Fabaceae infected with *Erysiphe* spp. in Australia.** Symptoms on (A) *Vigna radiata* BRIP 71005 infected with *Erysiphe vignae,* (B) *Acacia glaucoptera* BRIP 76682 infected with *Erysiphe* sp., (C) *Gastrolobium celsianum* BRIP 76684 infected with *Erysiphe guarinonii*, (D) *Acacia* sp. BRIP 76686 infected with *Erysiphe quercicola*, (E) *Acacia sophorae* BRIP 71600 infected with *Erysiphe quercicola*, and (F) *Cullen tenax* BRIP 76633 infected with *Erysiphe medicaginis*. Conidiophores of *Erysiphe medicaginis* (G), *Erysiphe quercicola* (H), and *Erysiphe vignae* (I) mounted in lactic acid. (J) Germinated and non-germinated conidium of *Erysiphe vignae* mounted in lactic acid. (K) Lobed, hyphal appressorium of *Erysiphe vignae*. Bar = 10 μm.

## Identification of powdery mildews on fabaceous hosts

Light microscopy observations of freshly collected powdery mildews and specimens rehydrated from herbarium materials morphologically identified all these pathogens as either *Erysiphe* or *Podosphaera* species. All specimens identified as *Erysiphe* had conidiophores that produced conidia singly; at least some of the hyphal appressoria were lobed; conidia did not contain fibrosin bodies; and when present, the position of the germ tubes that emerged from conidia was terminal or sub-terminal with simple or lobed ends (Figs 1-3).

In contrast, powdery mildews identified as *Podosphaera* always produced conidia on their conidiophores in chains; their hyphae had simple appressoria; fibrosin bodies were present in fresh conidia; germ tubes stemmed from the middle of conidia and terminated in simple apices (Fig 4).

BLASTn searches of the consensus ITS sequences determined in the fresh and herbarium specimens confirmed the morphological identifications at genus level. The sexual morphs known as chasmothecia were not found in any fresh or herbarium specimens examined in this study.

## Phylogenetic analyses

ITS sequences were determined for all 74 fresh and herbarium specimens examined in this study. The newly determined *Erysiphe* sequences were analysed together with 17 *Erysiphe* sequences published earlier from Australian fabaceous

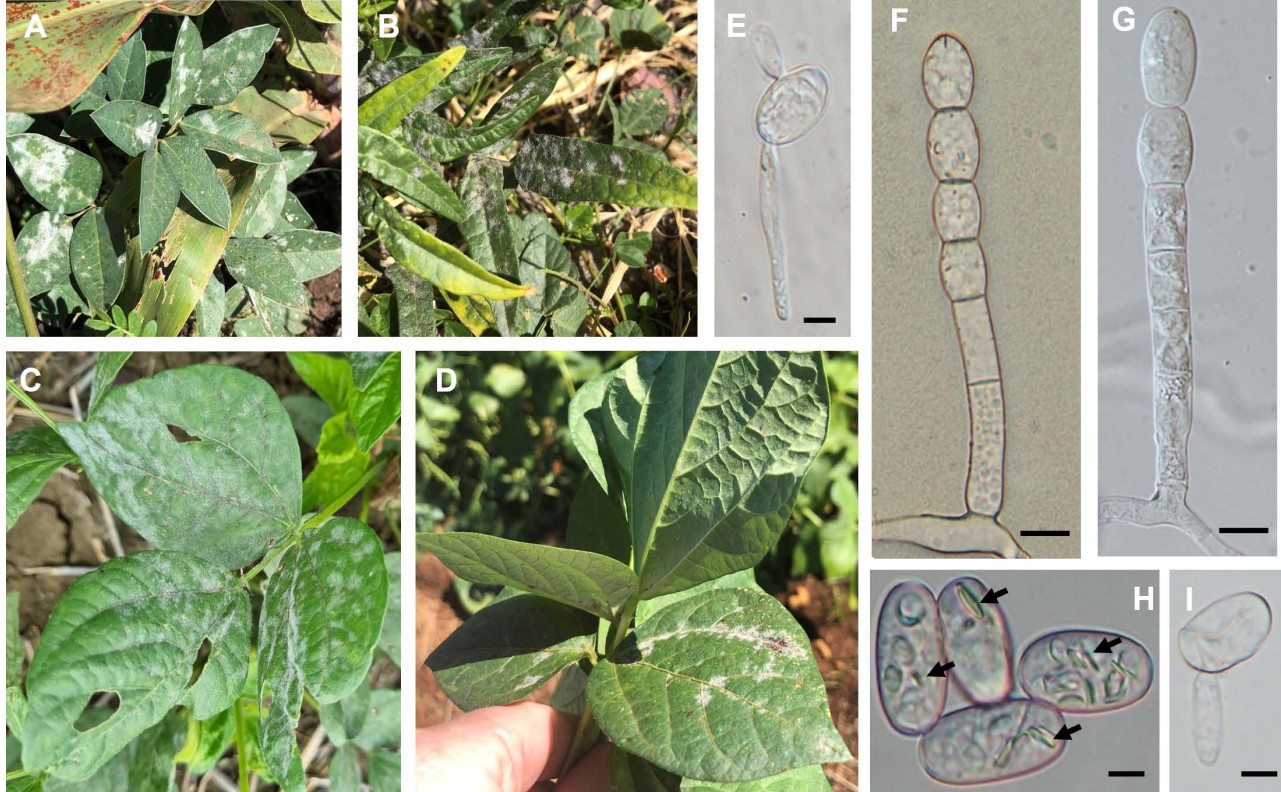

**Fig 4. Hosts in the Fabaceae naturally infected with *Podosphaera xanthii* in Australia.** (A) Symptoms on leaves of *Macroptilium lathyroides* BRIP 76642, (B) *Vigna lanceolata* BRIP 76624, (C) *Vigna radiata* BRIP 71599, and (D) *Vigna angularis* BRIP 76647. (E and I) Germinated conidia mounted in lactic acid. (F - G) Conidiophores mounted in lactic acid. (H) Conidia mounted in lactic acid. Arrows point to fibrosin bodies. Bar = 10 μm.

hosts [16–21,23,35] and four reference sequences [13,29,30,36], while the new *Podosphaera* sequences were included in a separate phylogenetic analysis that contained two previously published *Podosphaera* sequences from the Fabaceae in Australia [20] with eight reference sequences [16,19]. Altogether, 65 specimens were identified as *Erysiphe* and 28 as *Podosphaera* species on the Fabaceae in Australia (Table 3).

The analysis of the *Erysiphe* specimens revealed the occurrence of a total of ten species on 37 fabaceous hosts in Australia (Table 3 and Fig 5) while the other phylogeny indicated that all *Podosphaera* specimens from 18 hosts in the Fabaceae belong to a single species, *P. xanthii* (Table 3 and Fig 6). Altogether, eleven powdery mildew species were identified in Australia from more than 50 host species belonging to 24 genera of the Fabaceae (Table 2).

## Specimens belonging to the genus *Erysiphe*

Based on ITS sequences and microscopy, a total of 65 Fabaceae specimens examined in this study were naturally infected by powdery mildews belonging to the *Erysiphe* genus (Table 2; Table 3). In total, this study reports 28 new host records for *Erysiphe* species in Australia (Table 2). Ten species of *Erysiphe* were confirmed in this study, naturally occurring on 37 Fabaceae host species covering 17 host genera (Tables 2 and 3). Of the ten *Erysiphe* species, *E. diffusa* and *E.* cf. *trifoliorum*, were the most common, found on eleven and ten hosts, respectively (Table 3). Several other *Erysiphe* species were detected in this study on fewer hosts, such as *E. pisi* which was only found on a single host. Each of these species caused typical powdery mildew symptoms on the leaves, and sometimes phyllodes, of their hosts that included native Australian plants (Figs 1-3).

**Table 3. Summary of the powdery mildew species detected on fabaceous hosts in Australia.**

| Powdery mildew species[1] | Number of specimens | Number of host species | Number of host genera |
|---|---|---|---|
| *Erysiphe aquilegiae* | 3 | 2 | 2 |
| *E.* cf. *trifoliorum* | 18 | 10 | 7 |
| *E. diffusa* | 17 | 11 | 7 |
| *E. guarinonii* | 2 | 2 | 2 |
| *E. medicaginis* | 8 | 6 | 4 |
| *E. neolycopersici* | 2 | 2 | 2 |
| *E. pisi* | 4 | 1 | 1 |
| *E. quercicola* | 4 | 4 | 1 |
| *Erysiphe* sp. | 1 | 1 | 1 |
| *E. vignae* | 6 | 3 | 2 |
| *Podosphaera xanthii* | 28 | 18 | 10 |

[1]Identified based on morphology and ITS sequencing

*Erysiphe diffusa* was found on eleven hosts, including three species of *Glycine* (Table 2 and Fig 1). All samples of *E. diffusa* had similar morphology and identical ITS sequences, except specimen BRIP 68827 from *Glycine clandestina*, which differed from all other Australian specimens examined in this work by one nucleotide [16]. The ITS sequences of *E. diffusa* determined in this study were identical to several *E. diffusa* ITS sequences available in GenBank, including AB078800, reported as *E. diffusa* from soybean in Japan [29]. *Erysiphe diffusa* was detected on eleven Fabaceae species in this study, including several important agricultural crops, such as soybean, pigeonpea, and common bean (Table 2). Several Australian native Fabaceae hosts were naturally infected with *E. diffusa*, including *Cullen australasicum*, *G. tabacina*, and *G. clandestina*.

*Erysiphe* cf. *trifoliorum* was confirmed on ten Fabaceae host species, covering seven genera, in this study (Tables 3 and Fig 2). *Erysiphe* cf. *trifoliorum* was found on several important agricultural crops, including several species of *Vicia*, pea (*Pisum sativum*), and common bean (Table 2). Fifteen *E.* cf. *trifoliorum* specimens had identical ITS sequences, while BRIP 2236, BRIP 76640 and BRIP 76641 differed by up to two nucleotides. ITS sequences that are identical to those determined in fifteen specimens in this work are available in GenBank, including MN216308 from *Trifolium hybridum* in Korea, reported as *E. trifoliorum*, and FJ378877 from *Medicago* sp. in the USA, and reported as *E. trifolii*. The taxonomy of *E.* cf. *trifoliorum* remains unresolved [12,14,16].

*Erysiphe medicaginis* was recently described from Australia [18] and detected on six Fabaceae species in this study (Table 2 and Fig 3). *Medicago* species, which are grown as pasture crops and are also common weeds throughout Australia, were the most common hosts. Two Australian native species, *Cullen tenax* and *Hardenbergia comptoniana*, were also infected with *E. medicaginis*. *Erysiphe medicaginis* was detected in Queensland, New South Wales, South Australia, and Western Australia (Table 2). ITS sequences of *E. medicaginis* determined in this study were identical to several ITS sequences in GenBank, including LC009919 from *Sophora flavescens* in Japan, another Fabaceae host, and reported as a *Pseudoidium* species [37].

*Erysiphe quercicola* was confirmed on four Fabaceae species, all belonging to the *Acacia* genus (Table 2 and Fig 3). All *E. quercicola* ITS sequences in this study were identical to several ITS sequences on GenBank, such as OM033344 reported as *E. quercicola* from *Mangifera indica* in Taiwan [38]. Two powdery mildews on *Cassia fistula* and another one on *Arachis pintoi* were identified as *E. aquilegiae*. The ITS sequences on the two *C. fistula* specimens were identical and differed by one nucleotide from the *A. pintoi* specimen. Specimens identified as *E. neolycopersici* were detected on two Fabaceae species and grouped together within the *E. aquilegiae* clade (Table 2 and Fig 5). ITS sequences of the

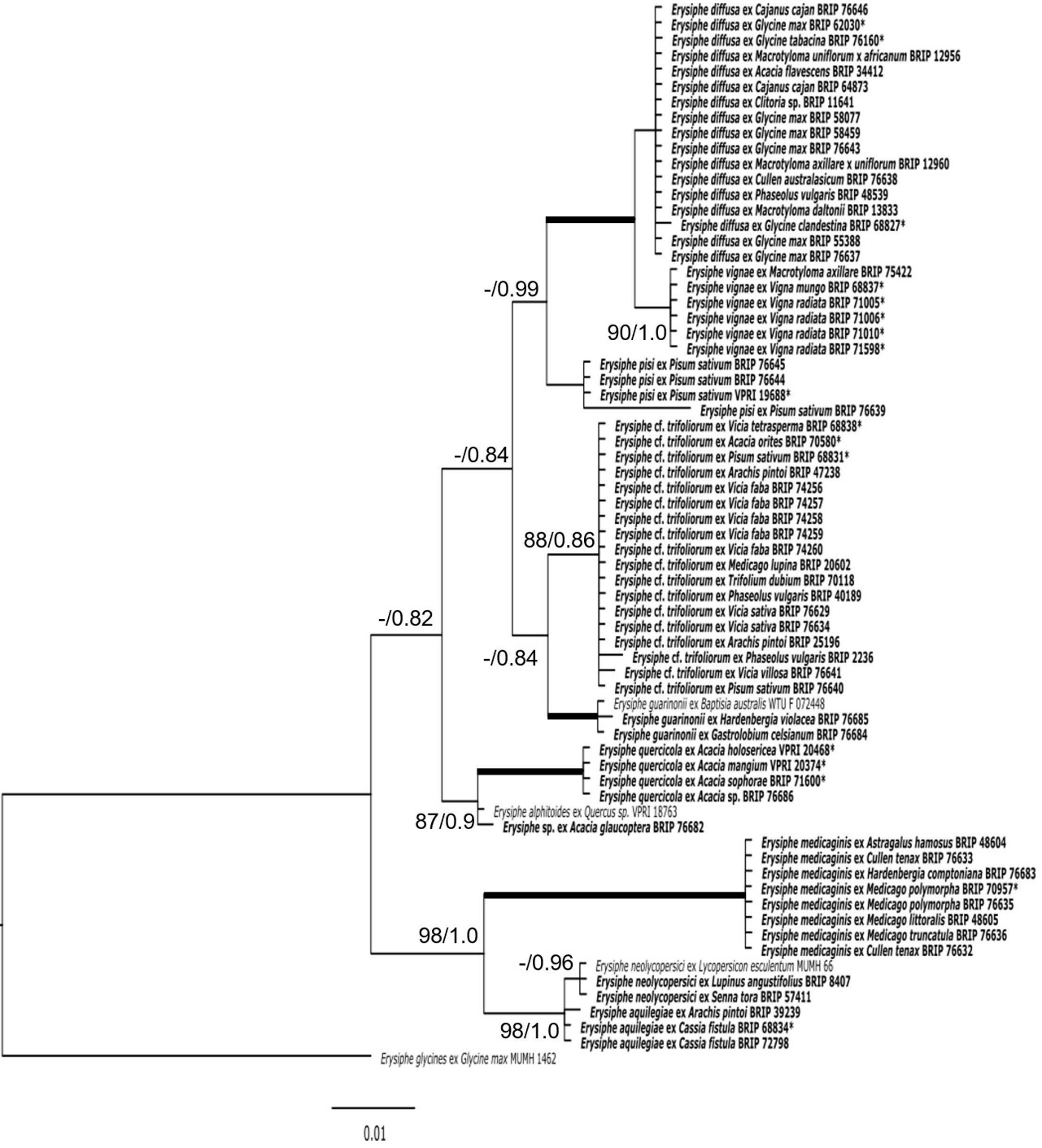

**Fig 5. Maximum likelihood phylogram based on the internal transcribed spacer (ITS) sequences of the nuclear ribosomal DNA (nrDNA) of powdery mildew specimens belonging to the genus *Erysiphe*.** The specimens collected from fabaceous hosts are in bold. Previously published sequences from Australia are followed by '*'. All other specimens were obtained from GenBank. The tree is rooted to the ITS sequence of *Erysiphe glycines* MUMH 1462. Maximum Likelihood bootstrap values >80% and Bayesian Posterior Probability values >0.80 are shown above or below the branches. Thickened branches represent Maximum Likelihood bootstrap values of 100% and Bayesian Posterior Probability of 1.00. The scale bar represents nucleotide substitutions per site.

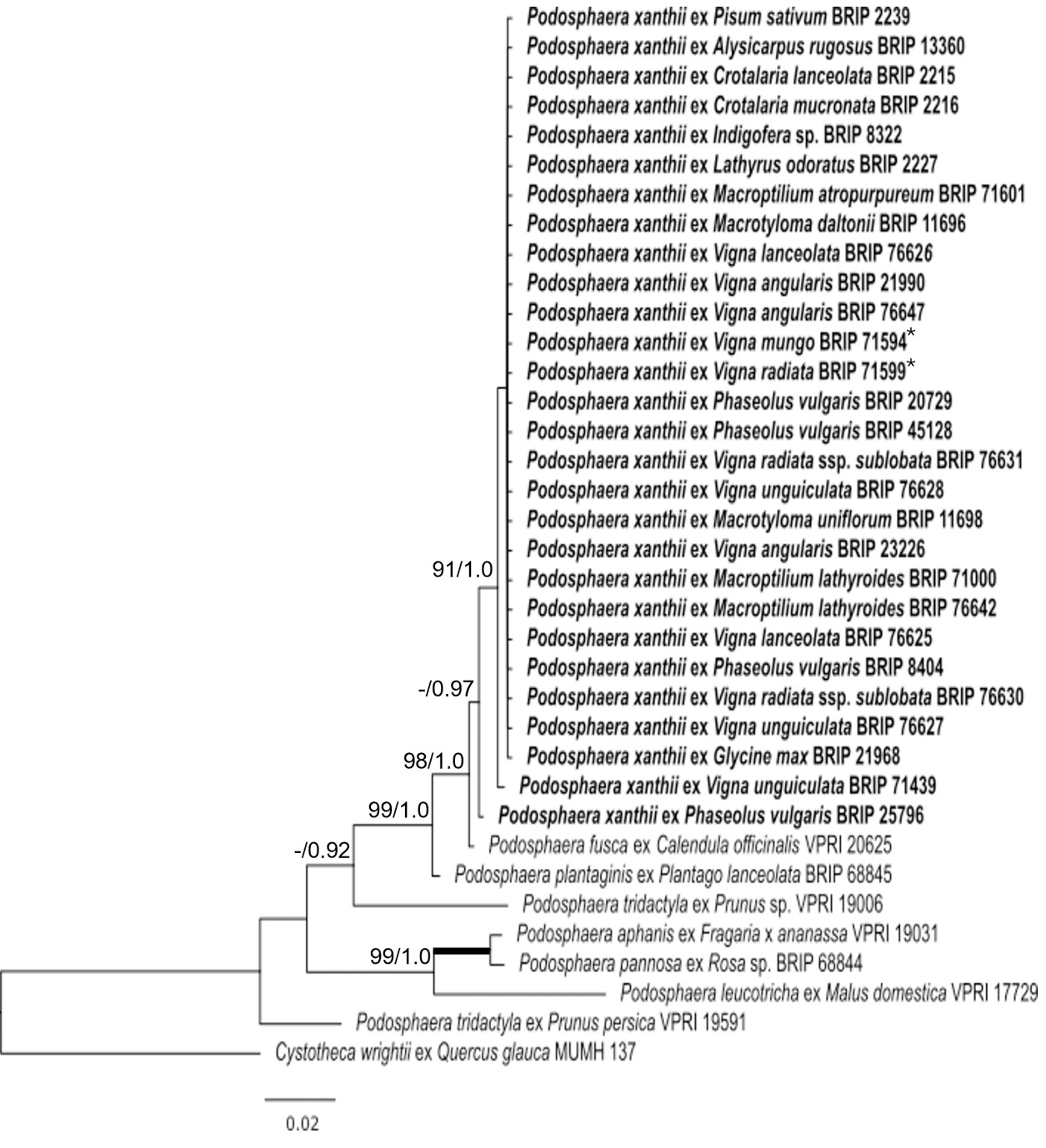

**Fig 6. Maximum likelihood phylogram based on the internal transcribed spacer (ITS) sequences of the nuclear ribosomal DNA (nrDNA) of powdery mildew specimens belonging to the genus *Podosphaera*.** The specimens collected from fabaceous hosts are in bold. Previously published sequences from Australia are followed by '\*'All other sequences were obtained from GenBank. The tree is rooted to the ITS sequence of *Cystotheca wrightii* MUMH 137. Maximum Likelihood bootstrap values >80% and Bayesian Posterior Probability values >0.80 are shown above the branches. Thickened branches represent Maximum Likelihood bootstrap values of 100% and Bayesian Posterior Probability of 1.00. The scale bar represents nucleotide substitutions per site.

*E. neolycopersici* specimens were identical and differed by one or two nucleotides to *E. aquilegiae*. *Erysiphe guarinonii* was found on two Fabaceae hosts, both native to Australia, *Hardenbergia violacea* and *Gastrolobium celsianum*. *Erysiphe pisi* was confirmed on pea and was the only pathogen detected on a single host in this study. One *Erysiphe* specimen, BRIP 76682, collected from *Acacia glaucoptera* in Western Australia differed by one nucleotide from AB292705, the ITS sequence of a powdery mildew identified earlier as *E. alphitoides* on *Quercus* sp. in Australia [35].

*Erysiphe vignae* described recently from mungbean and black gram [20], was detected on a new host in Australia in this study. The ITS sequence and morphology of the powdery mildew fungus on perennial horse gram (*Macrotyloma axillare*) confirmed its identity as *E. vignae* (Table 2 and Fig 5). A recent BLASTn search revealed several ITS sequences from overseas that are identical to *E. vignae*, including powdery mildew samples from *Phaseolus vulgaris* in Spain, deposited in GenBank as *Erysiphe* sp. (KU320678) and China, as *E. vignae* (MW579545); *P. acutifolius* in Puerto Rico, as *E. diffusa* (PP938951); *P. coccineus* in Mexico, as *Erysiphe* sp. (OQ448664); *V. unguiculata* in Brazil, as *E. diffusa* (KY515231) and China, as *E. vignae* (ON073844); *G. max* in China, as *E. diffusa* (MG171170); *Erythrina indica* in Brazil, as *Erysiphe* sp. (MF326644); and *Strophostyles pauciflora* in USA, as *E. vignae* (PP681083).

Of all the Fabaceae hosts included in this study, those within the Australian native *Acacia* genus were infected with the most diverse powdery mildew species. Four different species of *Erysiphe* were detected on the different *Acacia* host species, including *E. quercicola*, *E. diffusa*, *E.* cf. *trifoliorum*, and an *Erysiphe* sp. with an ITS sequence that most closely matched *E. alphitoides* (Table 2). More than one species of powdery mildew was recorded from many of the established crop hosts, including common bean, pea, soybean, mungbean, and black gram. Often these crops were hosts of both *P. xanthii* and one or more *Erysiphe* species.

### Specimens belonging to the genus *Podosphaera*

Twenty-eight Fabaceae specimens were confirmed to be infected with *Podosphaera xanthii* (Tables 2 and 3). *Podosphaera xanthii* was the most common powdery mildew that infected Fabaceae hosts in this study, consisting of 18 host species across ten genera (Table 3). This study reports 15 new host records for *P. xanthii* in Australia (Table 2). *Podosphaera xanthii* was most prevalent on the *Vigna* genus and was found on seven different species, including the Australian natives, wild mungbean (*V. radiata* ssp. *sublobata*) and maloga bean (*V. lanceolota*) (Table 2 and Fig 4). All but two *P. xanthii* sequences determined in this study were identical and grouped together within a single clade with a Bayesian posterior probability of 1.0 and a Maximum Likelihood bootstrap value of 91% (Fig 6). The ITS sequence of the *P. xanthii* specimen BRIP 71439 from *V. unguiculata* differed by one nucleotide, and BRIP 8404 from *P. vulgaris* by five nucleotides from the other *P. xanthii* sequences; however, these two specimens remained grouped together within the *P. xanthii* clade (Fig 6).

### Discussion

This study was triggered by a lack of knowledge on the host range of *Erysiphe vignae*, a species that was recently discovered on mungbean and black gram in Australia [20]. Morphological examination and ITS sequencing of 34 freshly collected field samples of powdery mildews infecting diverse Fabaceae species in Australia, and 40 Australian herbarium specimens collected from 1928, detected *E. vignae* on *M. axillare* in Queensland. *Macrotyloma axillare*, commonly known as perennial horse gram, is a heat and drought tolerant legume originating from sub-Saharan Africa and cultivated for livestock forage in subtropical and tropical regions of Australia [39–40]. Like many other legumes originally introduced from overseas as pasture crops [4], *M. axillare* has become an environmental weed in Queensland. It has a negative impact on the regeneration of native species by climbing on woody plants in open forests and woodlands [5].

BLASTn searches in GenBank revealed several powdery mildews with ITS sequences identical to *E. vignae*, which were collected outside Australia from hosts other than mungbean, black gram and *M. axillare*. These records from Brazil, China, Mexico, Spain, Puerto Rico, and the USA indicate that *E. vignae* may have a global distribution on diverse legume species and could be present in Australia on, as yet, unknown hosts.

Outside of Australia, *E. polygoni* was repeatedly reported as the causal agent of powdery mildew on *Vigna* spp. and other fabaceous hosts [41]. This binomial often refers to all powdery mildews on Fabaceae [12]. Interestingly, powdery mildews with ITS sequences that are highly similar to diverse *E. polygoni* records in GenBank were not detected in this study.

Molecular identification of powdery mildew species was solely based on ITS sequence analyses in this work. ITS and other nrDNA sequences have long been used as reliable molecular tools for powdery mildew identifications and phylogenies [37,42,43] and a genome-scale phylogeny of the Erysiphaceae based on 751 single-copy orthologs extracted from 24 powdery mildew genomes supported the lineages that were previously established based on nrDNA analyses [11]. Recent advances in the identification and phylogeny of different groups of the Erysiphaceae included analyses of a number of secondary DNA species barcodes, including fragments of genes for glyceraldehyde-3-phosphate dehydrogenase (*GAPDH*), calmodulin (*CAL*), glutamine synthetase synthase (*GS*), β-tubulin (*TUB2*), actin (*ACT*), and mini-chromosome maintenance protein 7 (*MCM7*) [36,44–47]. These multi-locus analyses focusing on species-level identifications highlighted that some of the secondary barcode sequences confirmed even single nucleotide differences in the ITS region of diverse specimens [36,45–47]. Therefore, ITS sequences were considered as sufficient to achieve the aims of the present study and even single nucleotide differences in this locus were noted during analyses.

Prior to this study, only eight species of powdery mildew had been identified on fabaceous hosts in Australia based on morphological characteristics and ITS analyses [15–23]. This study revealed the presence of *P. xanthii* and ten species of *Erysiphe* on more than 50 fabaceous hosts in Australia, including 17 native host species. Overall, this study reports 43 new host records for powdery mildews infecting the Fabaceae in Australia.

*Podosphaera xanthii* was detected more on fabaceous hosts than any other powdery mildew in this study. It is regarded as a species complex, comprised of multiple races, with distinct or broad host ranges [10,48,49]. This study supported its presence on mungbean, cowpea, wild mungbean (*V. radiata* ssp. *sublobata*), and black gram [16,20] and identified an additional 14 fabaceous hosts of *P. xanthii* in Australia. *Podosphaera xanthii* is a major pathogen of mungbean in Australia, occurring in all regions of production each year [20]. Two Australian native *Vigna* species were identified as hosts of *P. xanthii* in this study, namely wild mungbean and maloga bean (*V. lanceolata*). Both species are known to grow in close proximity to mungbean paddocks in Australia; therefore, may contribute to the survival of the pathogen between mungbean cropping seasons.

*Erysiphe diffusa* had the broadest host range of all *Erysiphe* spp. and was detected on eleven fabaceous species in this study. Previously, it was recorded on soybean and two Australian natives, *G. clandestina* and *G. tabacina* in Australia [16,17,20,22]. Those three species were confirmed as hosts in this study, in addition to *Acacia flavescens, Clitoria* sp., pigeonpea, common bean, various *Macrotyloma* spp., and the Australian native, *Cullen australasicum*. Although pigeonpea is currently only a minor crop in Australia, it has been identified as having a high production potential in Queensland and its industry is growing [50]. As its production expands throughout Queensland, it is possible that *E. diffusa* could become a significant constraint to its production.

A recent study by Kelly et al. [17] confirmed that *G. tabacina* is an alternate host for *E. diffusa* infecting soybean. It is likely that the other hosts identified in this study also act as alternate hosts for this crop pathogen. Outside of Australia, *E. glycines* has also been reported as a pathogen of soybean [29], however this species was not detected in this study.

*Erysiphe* cf. *trifoliorum* was the third most common powdery mildew pathogen of fabaceous hosts in Australia. It was found on ten host species across seven genera. The taxonomy of *E.* cf. *trifoliorum* remains unresolved; Bradshaw et al. [12] proposed that the *E.* cf. *trifoliorum* complex includes at least four separate species that are widely reported on the Fabaceae. Cunnington et al. [15] also reported *E.* cf. *trifoliorum* as the most common species of powdery mildew on the Fabaceae in Australia.

This study has reported several new hosts of *E. medicaginis* in Australia. *Erysiphe medicaginis* was only recently described as a new species in Australia, occurring on *M. polymorpha* [18]. Six fabaceous species were newly identified as

hosts of *E. medicaginis* in this study, mostly within the *Medicago* genera. Various *Medicago* species are grown as pasture crops throughout Australia and can be commonly found on roadsides as weeds. Two Australian native hosts, *Cullen tenax* and *Hardenbergia comptoniana*, were also infected with *E. medicaginis*. Milk vetch (*Astragalus hamosus*) was also found to host *E. medicaginis.*

*Erysiphe guarinonii* was identified on two fabaceous hosts, both native to Australia, *Hardenbergia violacea* and *Gastrolobium celsianum*. A comprehensive study by Bradshaw et al. [12] revealed that all hosts of *E. guarinonii* belong to the Fabaceae family. In our study, the ITS sequences of the powdery mildew on *H. violacea* (BRIP 76685) were identical to *Oidium hardenbergiae* in GenBank (AY450959). Cunnington et al. [15] had previously identified *O. hardenbergiae* on three *Hardenbergia* spp. in Australia, including *H. comptoniana*. In our study, we report *E. medicaginis* on *H. comptoniana.* Further specimens should be examined to determine whether *H. comptoniana* is a host of both species of powdery mildew.

Historically, *E. pisi* was considered to cause disease in most pulse crops and was made up of several *formae speciales* [3,51]. More recent studies suggest that *E. pisi* is likely confined to powdery mildew on *Pisum* [12]. In this work, *E. pisi* was only detected on pea, similar to a previous Australian study [15]. *Erysiphe* cf. *trifoliorum* was also detected on pea in this work and overseas [12,52].

Several crop pathogens were identified in this study on hosts that are not grown as commercial crops in Australia, indicating the opportunity for these to act as alternate hosts. Recently, ITS and *MCM7* sequencing and cross-inoculations confirmed that *G. tabacina*, and Australian native legume is a host of *E. diffusa* pathogenic to soybean [17]. The current work identified *M. axillare* as an alternate host of *E. vignae* infecting mungbean and black gram based on ITS sequences and morphology; and also, a number of new hosts of the crop pathogens, *P. xanthii* and *E. diffusa*. Successful cross-inoculation tests would be needed to support these results. However, it is often difficult to carry out such tests that require fresh inoculum, ideally from all potential host plant species, and disease-free host plants grown in isolation before and after inoculations with the respective powdery mildews [17,20]. Cross-inoculation tests were not included in this work.

No chasmothecia were observed on any host tissues during this study. Few powdery mildew species have been reported to develop chasmothecia in tropical and subtropical regions around the world [11]. It is likely that the Australian climate, and in some species, a broad host range, allows the continual survival of the asexual stage of powdery mildew species on multiple hosts throughout the year.

Lists of plant pathogens present in Australia are essential for biosecurity awareness and risk assessment. This study provides the most comprehensive catalogue of powdery mildew species infecting fabaceous hosts in Australia, providing insights into alternate hosts of key crop pathogens and strengthening Australia's plant biosecurity awareness. From a crop disease management perspective, the results highlight the importance of weed control in and around crop paddocks to reduce the sources of inoculum during crop production and limit the survival of powdery mildews on alternate hosts within and between cropping seasons.

## Acknowledgments

The authors would like to greatly acknowledge the support of the Australian Department of Agriculture, Fisheries and Forestry Brisbane Science and Surveillance team members, Jennifer Morrison, Dr Louisa Parkinson, and Jamie Summerhayes for the collection and identification of *Erysiphe vignae* on *Macrotyloma axillare*; Dr Trevor Volp and Adam Quade (Queensland Department of Primary Industries) for the collection of two specimens; and Drs Anke Martin (University of Southern Queensland), Andrew Taylor (Government of Western Australia, Department of Primary Industries and Regional Development) and Eden Tongson (The University of Melbourne) for the collection of one specimen. This research was supported by the University of Southern Queensland (UniSQ), and the Queensland Department of Primary Industries (DPI).

## Author contributions

**Conceptualization:** Lisa A Kelly, Levente Kiss.

**Data curation:** Lisa A Kelly, Levente Kiss.

**Formal analysis:** Lisa A Kelly, Niloofar Vaghefi.

**Investigation:** Lisa A Kelly, Buddhika A. Dahanayaka, Aftab Ahmad, Levente Kiss.

**Methodology:** Lisa A Kelly, Niloofar Vaghefi, Levente Kiss.

**Resources:** Lisa A Kelly, Levente Kiss.

**Supervision:** Niloofar Vaghefi, Levente Kiss.

**Visualization:** Lisa A Kelly.

**Writing – original draft:** Lisa A Kelly.

**Writing – review & editing:** Lisa A Kelly, Niloofar Vaghefi, Levente Kiss.

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
