## [Decision Letter · Decision Letter 0]

7 Mar 2025

PONE-D-24-54989An unexpected diversity of powdery mildew species infecting the Fabaceae in AustraliaPLOS ONE

Dear Dr. Kelly,

Thank you for submitting your manuscript to PLOS ONE. After careful consideration, we feel that it has merit but does not fully meet PLOS ONE’s publication criteria as it currently stands. Therefore, we invite you to submit a revised version of the manuscript that addresses the points raised during the review process.

We look forward to receiving your revised manuscript.

Kind regards,

Abhay K. Pandey

Academic Editor

PLOS ONE

Journal Requirements:

Additional Editor Comments:

The MS is well organized and written—indetails; however some minor corrections are required before acceptance. I have mentioned some comments on the MS that authors can find useful in revision.

Reviewers' comments:

Reviewer's Responses to Questions

**Comments to the Author**

1. Is the manuscript technically sound, and do the data support the conclusions?

Reviewer #1: Yes

2. Has the statistical analysis been performed appropriately and rigorously? 

Reviewer #1: Yes

3. Have the authors made all data underlying the findings in their manuscript fully available?

Reviewer #1: Yes

4. Is the manuscript presented in an intelligible fashion and written in standard English?

Reviewer #1: Yes

5. Review Comments to the Author

Please use the space provided to explain your answers to the questions above. You may also include additional comments for the author, including concerns about dual publication, research ethics, or publication ethics. (Please upload your review as an attachment if it exceeds 20,000 characters.

Reviewer #1: Line 63: I would consider adding a few sentences describing the life cycle of powdery mildews on the fabaceae, if a more or less general life cycle is available. This will help the uniformed reader to better understand the importance of the sexual stage, and also why alternative hosts of powdery mildew can be important contributors to disease (lines 82-86). Also include the importance of heterothallic species causing powdery mildew in terms of development of new genotypes of the pathogen(s).

Line 93: move comma after "including" to after "general".

Line 105: Delete "crop"

Line 126: A concern here is if single-conidium isolates was obtained. No mention is made if this was done and if so, how. It is, however, important to obtain single-conidium isolates (different methodologies have been described in literature, given that it is an obligate pathogen), but it’s the only way to ensure that species identification was done using genetically pure specimens.

Line 241: Why was no chasmothecia observed? Is it an inherent trait of these species that chasmothecia is not readily produced, or was it due to samples being collected prior to formation of chasmothecia? Please add a line of two in the discussion on this topic.

Line 245: The no of specimens is confusing. Mention is made of 34 fresh specimens in line 171, and when you add up the rest of the specimens (line 171-177) it equals 69 (34 + 17 + 9 + 9). Please clarify.

Line 254: Mention as a subscript below the table that the species was identified molecularly, so readers won’t be confused about how identification was done.

Line 428: Add a fullstop at the end of the sentence.

6. PLOS authors have the option to publish the peer review history of their article (what does this mean? ). If published, this will include your full peer review and any attached files.

**Do you want your identity to be public for this peer review?** For information about this choice, including consent withdrawal, please see our Privacy Policy .

Reviewer #1: No

[NOTE: If reviewer comments were submitted as an attachment file, they will be attached to this email and accessible via the submission site. Please log into your account, locate the manuscript record, and check for the action link "View Attachments.". If this link does not appear, there are no attachment files.

While revising your submission, please upload your figure files to the Preflight Analysis and Conversion Engine (PACE) digital diagnostic tool, https://pacev2.apexcovantage.com/ . PACE helps ensure that figures meet PLOS requirements. To use PACE, you must first register as a user. Registration is free. Then, login and navigate to the UPLOAD tab, where you will find detailed instructions on how to use the tool. If you encounter any issues or have any questions when using PACE, please email PLOS at figures@plos.org . Please note that supporting information files do not need this step.

---

## [Author Response · Author response to Decision Letter 1]

24 Mar 2025

Reviewer’s comments to the authors:

Author response: The manuscript meets PLOS ONE’s style requirements.

Author response: No permits were required to undertake this study and complied with all relevant regulations.

Author response: To our knowledge, all references listed are complete and correct.

Additional Editor Comments:

The MS is well organized and written—indetails; however some minor corrections are required before acceptance. I have mentioned some comments on the MS that authors can find useful in revision.

Author response: Thank you! We have incorporated almost all suggestions and corrections in the revised manuscript.

The number of Fabaceae species has been clarified in the Abstract on line 27. Additional sentences were added to the Abstract in lines 27 to 30 to provide further detail on the results of the study.

In regards to the number of genera and species that comprise the Fabaceae on line 35, we have kept the original numbers as “640 genera and 18000 species” as that is what is listed in the cited reference (https://www.sciencedirect.com/science/article/abs/pii/B0121451607001988?via%3Dihub).

Forage legumes have been added to line 38.

The authority name for each plant species has been listed in lines 41 to 44.

Reviewers' comments:

Reviewer #1: Line 63: I would consider adding a few sentences describing the life cycle of powdery mildews on the fabaceae, if a more or less general life cycle is available. This will help the uniformed reader to better understand the importance of the sexual stage, and also why alternative hosts of powdery mildew can be important contributors to disease (lines 82-86). Also include the importance of heterothallic species causing powdery mildew in terms of development of new genotypes of the pathogen(s).

Author response: As suggested by the reviewer, we have added a brief description of the asexual and sexual powdery mildew life cycles on lines 63-72.

Line 93: move comma after "including" to after "general".

Author response: Done. The comma has now been moved to after “general” on line 108.

Line 105: Delete "crop"

Author response: As suggested, we have removed the word, “crop” on line 120.

Line 126: A concern here is if single-conidium isolates was obtained. No mention is made if this was done and if so, how. It is, however, important to obtain single-conidium isolates (different methodologies have been described in literature, given that it is an obligate pathogen), but it’s the only way to ensure that species identification was done using genetically pure specimens.

Author response: Our studies, just like many other similar works (e.g., those cited in the manuscript: see references nos. 12-23; 29-30; 35-38; and 44-49) were based on powdery mildew samples freshly collected from the field or preserved as herbarium specimens in internationally recognised collections.

Species identifications were based on these field or herbarium samples (i.e., populations of the respective powdery mildew fungi) based on (a) morphological characteristics that readily distinguish Podosphaera and Erysiphe species (see Figs. 1-4 and the text); and (b) ITS sequences that are always different in different powdery mildew species as revealed, for example in the papers cited in the manuscript and listed above.

We are not aware of a single publication that used single single-conidium isolates for the identification of powdery mildew species. This is simply not needed for this purpose and would have been impossible for example in the case of the dried (dead) herbarium specimens used in this work and other publications mentioned above. There is no indication in the literature that morphological patterns and/or ITS sequences would differ amongst single-conidium isolates of the same species. Intra-specific variation of ITS sequences was detected in a few powdery mildew samples as “exceptions to the rule” (this was done by our group: Kovacs et al. 2011, Eur J Plant Pathol 131: 135-141; and Kiss 2012, PNAS 109: E1811). However, these rare intra-specific differences were detected amongst powdery mildew populations, not single-conidium isolates.

Single-conidium isolates are regularly produced in our laboratory for whole-genome sequencing projects and also when we study for example mating types or fungicide resistance patterns in diverse powdery mildews. In those studies, single-conidium isolates are the standard way of conducting research, whereas species identifications do not require this approach.

Line 241: Why was no chasmothecia observed? Is it an inherent trait of these species that chasmothecia is not readily produced, or was it due to samples being collected prior to formation of chasmothecia? Please add a line of two in the discussion on this topic.

Author response: This is an interesting question. Our study supports earlier findings that most powdery mildews do not produce chasmothecia in Australia (see references 15-17 and 19, 20). We propose that the Australian climate allows the continual survival of the asexual stage of most powdery mildew species on their hosts. We have added a few sentences to the discussion on lines 498 to 502.

Line 245: The no of specimens is confusing. Mention is made of 34 fresh specimens in line 171, and when you add up the rest of the specimens (line 171-177) it equals 69 (34 + 17 + 9 + 9). Please clarify.

Author response: Thank you for your suggestion. To clarify, there were 34 freshly collected powdery mildew specimens and 40 herbarium specimens as stated in lines 186-188. An additional 19 sequences from recently identified species from Fabaceae hosts were included in the phylogenetic analyses as references: 17 for Erysiphe and 2 for Podosphaera (Figs. 5 and 6). Of the fresh and herbarium specimens, 17 specimens belonged withing the Vigna genus, 9 within the Vicia genus, and 9 within the Glycine genus. Specimens from other genera were not mentioned in the text but are shown in the Tables. All numbers referring to the samples/specimens included in the different parts of this study were double checked in the manuscript. Also, the number of samples/specimens was specified in the revised Abstract (lines 27-30).

Line 254: Mention as a subscript below the table that the species was identified molecularly, so readers won’t be confused about how identification was done.

Author response: Great suggestion. I have added the subscript to the table to clarify how the species were identified.

Line 428: Add a fullstop at the end of the sentence.

Author response: Done (line 443).

---

## [Decision Letter · Decision Letter 1]

2 Apr 2025

PONE-D-24-54989R1An unexpected diversity of powdery mildew species infecting the Fabaceae in AustraliaPLOS ONE

Dear Dr. Kelly,

Thank you for submitting your manuscript to PLOS ONE. After careful consideration, we feel that it has merit but does not fully meet PLOS ONE’s publication criteria as it currently stands. Therefore, we invite you to submit a revised version of the manuscript that addresses the points raised during the review process.

We look forward to receiving your revised manuscript.

Kind regards,

Abhay K. Pandey

Academic Editor

PLOS ONE

Journal Requirements:

Additional Editor Comments:

Please edit the references as per Journal Guideline

Reviewers' comments:

Reviewer's Responses to Questions

**Comments to the Author**

1. If the authors have adequately addressed your comments raised in a previous round of review and you feel that this manuscript is now acceptable for publication, you may indicate that here to bypass the “Comments to the Author” section, enter your conflict of interest statement in the “Confidential to Editor” section, and submit your "Accept" recommendation.

Reviewer #1: (No Response)

2. Is the manuscript technically sound, and do the data support the conclusions?

Reviewer #1: Yes

3. Has the statistical analysis been performed appropriately and rigorously? 

Reviewer #1: Yes

4. Have the authors made all data underlying the findings in their manuscript fully available?

Reviewer #1: Yes

5. Is the manuscript presented in an intelligible fashion and written in standard English?

Reviewer #1: Yes

6. Review Comments to the Author

Reviewer #1: Dear authors

Thank you for a well-written manuscript detailing excellent research to shed light on the status of powdery mildew species occurring on the Fabaceae in Australia. I hope future research can elucidate the identity of E. cf. trifoliorum, while future research on the origin of powdery mildews on introduced members of the Fabaceae (fungal species occurring only on the introduced Fabaceae and not the native species) can also be valuable.

One last edit that must be addressed is the References cited in Table 2. The publications cited are referred to by the author's name, and not according to the no in the References. Please correct.

One last small detail you can pay attention to is in line 134, where there's two spaces between "acid" and "on".

All other edits / suggestions have been met satisfactorily.

Best wishes for your future endeavors.

7. PLOS authors have the option to publish the peer review history of their article (what does this mean? ). If published, this will include your full peer review and any attached files.

**Do you want your identity to be public for this peer review?** For information about this choice, including consent withdrawal, please see our Privacy Policy .

Reviewer #1: No

---

## [Author Response · Author response to Decision Letter 2]

7 Apr 2025

Reviewers' comments:

1. Please note that your Data Availability Statement is currently missing the DOI/accession number of each dataset OR a direct link to access each database. If your manuscript is accepted for publication, you will be asked to provide these details on a very short timeline. We therefore suggest that you provide this information now, though we will not hold up the peer review process if you are unable.

Author response: All molecular data produced in the study is publicly available on GenBank. We have added the link to the GenBank database in the additional information. All collected materials were deposited in the Brisbane Plant Pathology Herbarium.

Author response: Australia is not currently a party to the Nagoya Protocol (https://www.dcceew.gov.au/science-research/australias-biological-resources/nagoya-protocol-convention-biological. We have updated the methods to include “No materials were collected from natural settings that require a collection permit” on line 120.

---

## [Editor Report · Decision Letter 2]

9 Apr 2025

An unexpected diversity of powdery mildew species infecting the Fabaceae in Australia

PONE-D-24-54989R2

Dear Dr. Kelly,

We’re pleased to inform you that your manuscript has been judged scientifically suitable for publication and will be formally accepted for publication once it meets all outstanding technical requirements.

Kind regards,

Abhay K. Pandey

Academic Editor

PLOS ONE
---

## [Editor Report · Acceptance letter]

PONE-D-24-54989R2

PLOS ONE

Dear Dr. Kelly,

I'm pleased to inform you that your manuscript has been deemed suitable for publication in PLOS ONE. Congratulations! Your manuscript is now being handed over to our production team.

Kind regards,

on behalf of

Dr. Abhay K. Pandey

Academic Editor

PLOS ONE